# An Efficient Vision Mamba–Transformer Hybrid Architecture for Abdominal Multi-Organ Image Segmentation

**DOI:** 10.3390/s25216785

**Published:** 2025-11-06

**Authors:** Fang Lu, Jingyu Xu, Qinxiu Sun, Qiong Lou

**Affiliations:** School of Science, Zhejiang University of Science and Technology, No. 318 Liuhe Road, Hangzhou 310023, China; lufang@zust.edu.cn (F.L.); 222309252008@zust.edu.cn (J.X.); 112012@zust.edu.cn (Q.S.)

**Keywords:** medical image segmentation, abdominal multi-organ, transformer, mamba, loss function

## Abstract

Accurate abdominal multi-organ segmentation is essential for disease diagnosis and treatment planning. Although numerous deep-learning models have been proposed, current methods still struggle to balance segmentation accuracy with computational efficiency, particularly for images exhibiting inhomogeneous intensity distributions and complex anatomical structures. To address these challenges, we present a hybrid framework that integrates an Efficient Vision Mamba (EViM) module into a Transformer-based encoder. The EViM module leverages hidden-state mixer-based state-space duality to enable efficient global context modelling and channel-wise interactions. In addition, a weighted combination of cross-entropy and Jaccard loss is employed to improve boundary delineation. Experimental results on the Synapse dataset demonstrate that the proposed model achieves an average Dice score of 82.67% and an HD95 of 16.36 mm, outperforming current state-of-the-art methods. Further validation on the ACDC cardiac MR dataset confirms the generalizability of our approach across imaging modalities. The results indicate that the proposed framework achieves high segmentation accuracy while effectively integrating global and local information, offering a practical and robust solution for clinical abdominal multi-organ segmentation.

## 1. Introduction

Abdominal multi-organ segmentation is critical for clinical diagnosis, surgical navigation, and disease treatment monitoring. With the rapid advancement of medical imaging technologies such as computed tomography (CT) and magnetic resonance imaging (MRI), the availability of imaging data has greatly increased, along with higher clinical demands for segmentation accuracy. Nevertheless, accurate and efficient segmentation remains challenging due to organ complexity, large inter-organ variability, and ambiguous boundaries. Existing methods still struggle to balance segmentation accuracy and model complexity, particularly when dealing with images characterized by uneven grayscale distribution and intricate anatomical structures.

Traditional abdominal multi-organ segmentation mainly relies on manual annotation by radiologists, which is not only inefficient, but also susceptible to inconsistent results due to subjective factors and differences in doctors’ experience [1]. To address these limitations, early machine learning methods such as support vector machines [2] and random forests [3] were introduced. While they showed promise in certain scenarios, they often struggled to handle the nonlinear and complex features of medical images.

With the rise of deep learning, automated abdominal multi-organ segmentation has gradually become the mainstream solution, improving diagnostic efficiency while ensuring reproducibility of results. Currently, medical image segmentation methods can be mainly classified into three categories: convolutional neural network (CNN)-based methods, Transformer-based methods, and the recently emerged Mamba methods based on state space models (SSMs). Among them, CNNs are popular for their efficient local feature extraction capability. The classical U-Net [4] adopts a CNN-based encoding-decoding structure with a jump-connection mechanism, which is able to capture image features efficiently and has become the basis of many subsequent CNN models. The models derived from this foundation, such as UNet++ [5], Residual UNet [6] and AttenUNet [7], achieve more significant segmentation performance improvement by further optimizing the feature extraction module and enhancing the feature fusion capability. In particular, Sharp U-Net [8] employs depthwise convolution with a sharpening kernel on encoder feature maps for binary and multi-class biomedical image segmentation. MEW-UNet [9] and MT-UNet [10] further integrate mechanisms like multi-axis frequency modeling and cross-task skip connections. However, CNNs inherently suffer from limited receptive fields, restricting their ability to effectively capture global contextual information, particularly in large-scale or structurally complex tasks.

To overcome this limitation, researchers have gradually introduced Vision Transformer (ViT) [11] to medical image segmentation tasks. TransUNet [12] first integrated ViT in CNN architecture, which significantly enhanced the model’s ability to model long-distance dependencies. Subsequently, Swin-UNet [13] and SwinUETR [14] introduced Swin Transformer [15] to reduce the computational cost and enhance the multi-scale feature extraction capability through the hierarchical structure and sliding window mechanism. DA-TransUNet [16] further embeds the dual-attention mechanism into the encoder and skip connection. It optimizes feature representation in both spatial and channel dimensions at the same time, which improves the model’s semantic understanding. Considering the complementary strengths of CNNs for local feature modeling and Transformers for capturing global dependencies, many studies have combined the two. Hybrid architectures, including TransNorm [17], FMD-TransUNet [18] and SwinPA-Net [19], have empirically shown improved performance in medical image segmentation by leveraging both local and global information. Therefore, fully leveraging the strengths of both models is essential. However, Transformer-based methods still exhibit certain limitations, including high computational complexity and memory demands.

Recently, SSMs [20] have attracted attention in medical image analysis due to their ability to model long sequences with linear computational complexity. Among them, Mamba, a structured state-space model [21,22], efficiently captures long-range dependencies and has become an active research direction. VM-UNet [23] was the first to apply Mamba to medical image segmentation, demonstrating strong performance across multiple public datasets. VMamba [24] introduced the Cross-Scan Module (CSM) to enhance hierarchical feature extraction, while U-Mamba [25] incorporated convolution into the SSM framework, combining local and global modeling. EfficientViM [26] further proposed a Hidden State Mixer-based State Space Duality (HSM-SSD) to enable efficient channel interactions in low-dimensional hidden space, reducing complexity while improving feature representation. These studies highlight the growing potential of SSM-based approaches in medical image segmentation, particularly for long-range dependency modeling.

While Mamba-based models provide efficient long-range dependency modeling, Mamba-based models still lag behind Transformers in capturing boundary-sensitive and semantically rich details [27]. Moreover, existing Mamba architectures require further improvement in terms of stability and generalization ability. Consequently, the three dominant methods remain complementary: CNNs excel at encoding local details but struggle to capture global interactions efficiently; Transformers provide rich semantic context and precise boundaries but suffer from high computational complexity and memory demands; and Mamba variants achieve linear complexity and long-range modeling but sacrifice some voxel-level accuracy. Motivated by this observation, we propose a hybrid architecture that combines the complementary advantages of Transformer and Mamba. Specifically, a lightweight Efficient Visual Mamba (EViM) module is introduced before the Transformer encoder. This module employs depthwise separable convolution to extract local context and uses the HSM-SSD to perform global channel interactions in a low-dimensional hidden space. A feed-forward network is then used to restore multi-scale semantic features. This design enhances global modeling without significantly increasing computational burden, and preserves local details that are crucial for accurate boundary delineation. Moreover, considering the characteristics of abdominal multi-organ segmentation, we systematically evaluate several region-level loss functions and adopt a weighted combination of Jaccard loss and cross-entropy loss, which improves segmentation accuracy and enhances cross-dataset generalization.

The main contributions of this research are summarized as follows:1.We integrate an EViM module before Transformer layers to preserve local feature details while enhancing the aggregation of global information and channel-wise interactions.2.We systematically compared three region-level loss functions—Dice, Jaccard, and Tversky. Jaccard loss yields the best convergence and segmentation performance.3.Extensive experiments on Synapse and ACDC datasets demonstrate state-of-the-art accuracy and strong cross-modal generalization.

## 2. Related Work

### 2.1. Transformer

The Transformer architecture has attracted considerable attention in medical image segmentation due to its strong capability in modeling long-range dependencies. As the first model to introduce a pure Transformer architecture into vision tasks, ViT provides a novel structural foundation for medical image analysis and has inspired a series of Transformer-based methods in this field. TransUNet [12] is one of the earliest models to combine Transformer with CNNs. It embeds ViT in the encoder for global context modeling and has achieved promising results in medical image segmentation. TransNorm [17] further integrates the Transformer module into both the encoder and skip connection paths of a standard U-Net. It also introduces a spatial normalization mechanism to adaptively recalibrate feature representations within skip connections. Swin-UNet [13], built upon the Swin Transformer [15], constructs a U-shaped architecture. By employing a hierarchical design and shifted window self-attention, it effectively reduces computational complexity and improves the fusion of local and global information. TransDeepLab [28], inspired by the DeepLabv3 architecture, builds a pure Transformer-based semantic segmentation network using hierarchical Swin Transformers, and introduces an atrous spatial pyramid pooling module to strengthen multi-scale feature modeling. FMD-TransUNet [18] further integrates a multi-axis external weight block (MEWB) and an enhanced dual-attention module (DA+) into the Transformer-based framework for precise abdominal multi-organ segmentation. Despite the progress made in global modeling and feature representation, current Transformer-based approaches still suffer from insufficient local detail modeling and inefficient channel information extraction. These limitations are particularly evident in complex tasks such as abdominal multi-organ segmentation. To better balance efficiency and accuracy, we incorporate a more efficient Mamba module into the Transformer-based encoder, enhancing channel information integration while reducing redundant features.

### 2.2. Mamba

With the rapid development of SSMs [20] in vision tasks, Mamba-based [21] approaches have gained increasing attention in medical image segmentation due to their efficient modeling capabilities and strong ability to capture long-range dependencies. VM-UNet [20] was the first model to introduce Vision Mamba [29] into the U-Net architecture, serving as an early exploration of SSMs in medical image segmentation. Building on this foundation, VMamba [24] proposed the CSM and a hierarchical structural design to enhance spatial contextual interaction. Given the high-resolution requirements of medical image segmentation, U-Mamba [25] incorporated convolutional operations into the SSM framework. This integration combines the long-range dependency modeling of SSMs with the local feature extraction capability of convolution, thus yielding improved capture of fine anatomical structures in medical images. Subsequently, enhanced models such as SegMamba [30] and SwinUMamba [31] were proposed, which integrated Visual Mamba with multi-scale fusion mechanisms or the Swin Transformer architecture to further improve segmentation performance. To enhance efficiency, EfficientViM [26] introduced an HSM-SSD architecture, which effectively captures global dependencies while significantly reducing computational complexity. However, despite their efficiency, current Mamba-based methods generally lag behind recent ViT-based models in terms of segmentation accuracy, particularly in scenarios involving blurred organ boundaries or significant structural variability. To address this limitation, we propose a Transformer-Mamba hybrid architecture by integrating a lightweight EViM module into the encoder of a Transformer-based model. This design leverages the strong feature modeling capability of the Transformer and the efficient channel interaction of EViM. The resulting architecture balances segmentation accuracy and inference efficiency, offering a more effective solution for medical image segmentation involving complex anatomical structures.

### 2.3. Dual Attention (DA) Mechanism

Although both Transformer- and Mamba-based methods enhance long-range modeling, they still under-represent local boundary details. Hence, attention mechanisms are incorporated to refine spatial and channel features. Attention mechanisms have been widely applied in medical image segmentation tasks to enhance the model’s ability to focus on key regions and informative features. The DA mechanism [32], which combines spatial and channel attention, has demonstrated strong performance in modeling global dependencies across various tasks. A representative example is DANet [33], which introduces separate spatial and channel attention modules to capture semantic dependencies among pixels, significantly improving scene segmentation accuracy. In the medical domain, SwinPA-Net [19] proposes the LPA module that fuses global–local attention for multi-scale context aggregation, thereby improving the recognition of organ locations and boundaries. D-SAT [34] introduces a dual-branch architecture that integrates dual semantic aggregation and dual attention modules, effectively enhancing performance in abdominal multi-organ segmentation. DA-TransUNet [16] incorporates the dual attention mechanism into both the encoder and skip connections, jointly optimizing feature extraction in spatial and channel dimensions, and further strengthening the representation of semantic structures in medical images. Although the above methods improve the model performance to a certain extent, the dual attention mechanism can capture long-range dependencies. However, it explicitly assigns weights to each channel through channel attention, lacks dedicated processing of neighborhood information, and relies on subsequent convolutional layers to compensate for local details. To address this limitation, we introduce the EViM module before the Transformer layers in the encoder. EViM integrates local convolution and global channel fusion into a single efficient module. Its core component, the HSM-SSD block, first employs depthwise separable convolution to extract neighborhood information, then performs global channel mixing in the hidden state space, and finally uses a feed-forward network to recover multi-scale details. This forward process effectively combines local texture and global semantics, achieving efficient global channel interaction.

## 3. Method

In this section, we will first introduce the overall structure of our model, then describe the roles and logical details of its important modules, and finally discuss the selection of the loss function.

### 3.1. Overall Framework

The overall network adopts a symmetric U-shaped architecture composed of a hybrid encoder, CNN-based decoder, and multi-scale skip connections, with the EViM module integrated to enhance the modeling capability. The overall structure is illustrated in Figure 1.

The encoder is built upon CNNs and is designed to progressively reduce spatial resolution while extracting multi-scale low-level semantic features. Given an input image $X\in R^{H\times W\times C}$, three consecutive convolutional blocks progressively down-sample the feature maps from H×W×16 to H/2×W/2×64, then to H/4×W/4×128 and finally to H/8×W/8×256. Each convolutional block is followed by a Multi-axis External Weight Block (MEWB), which leverages multi-directional frequency-domain modeling to capture joint local–global features and enhance feature expressiveness. The MEWB module, as a component of the baseline architecture FMD-TransUNet [18], is retained in our framework to ensure fair comparison and result consistency. The output of the MEWB is fed into the EViM module, which performs efficient channel-wise interactions and captures long-range dependencies through a state-space duality mechanism in a low-dimensional hidden space. The resulting feature map is then flattened in the embedding layer, where each token is projected into a *D*-dimensional embedding vector (with D=768), and subsequently processed by the Transformer encoder for deep semantic modeling. Finally, the Transformer output is reshaped to (768,H/16,W/16), passed through a 3×3 convolution with ReLU to produce (512,H/16,W/16), and introduced into the decoder. The three DA+ modules in the skip connections integrate the encoder features at the 1/2, 1/4, and 1/8 scales with the corresponding decoder features. The decoder is composed of convolutional blocks, cascaded upsampling operations, feature fusion modules, and a segmentation head. Each cascaded upsampling stage consists of two upsampling operators followed by a 3×3 convolution and a ReLU function. Through three cascaded upsampling stages, the feature maps are progressively transformed from (512,H/16,W/16) to (64,H/2,W/2). The resulting features then undergo a final upsampling step, and the segmentation head restores them to the original image size (H,W,16), producing accurate multi-organ predictions.

### 3.2. EViM Module

To enhance feature representation capability, we introduce the EViM module prior to the Transformer layers in the encoder. This is a deliberate design motivated by the need to preprocess the features. The EViM module is constructed based on the HSM-SSD layer, aiming to achieve efficient global information modeling and inter-channel interaction. Since the Transformer layers are sensitive to local information and redundancy in input features, the EViM module reduces this redundancy through its lightweight design [21]. By doing so, it directly enhances the Transformer’s ability to capture long-range dependencies, which is crucial for achieving higher segmentation accuracy. The architecture of this module is illustrated in Figure 2a.

The module consists of five main components: two Depthwise Separable Convolutions (DWConv) [35], Layer Normalization (LN) [36], the HSM-SSD layer, a Feed-Forward Network (FFN), and Batch Normalization (BN) [37]. Specifically, the DWConv employs 3×3 kernels and is designed to efficiently capture local neighborhood features. It is used before each FFN layer to enhance local perception. The FFN is composed of two consecutive 1×1 convolutions (pointwise convolutions) with a channel expansion ratio of 4, enabling cross-channel information interaction and nonlinear transformations. Specifically, the first convolution expands the channel dimension from *D* to 4D, followed by BN and ReLU activation to enhance nonlinear feature representation, while the second convolution projects the channel dimension back to *D* with zero-initialized normalization to ensure numerical stability during the early stage of training. The adoption of a 4× expansion ratio follows the standard configuration used in most Transformer- and Mamba-based visual backbones, such as ViT [11], Swin Transformer [15], and Vision Mamba [29]. At the normalization level, the EfficientViM [26] paper notes that although using BN for all operations is simple and efficient, this approach can lead to numerical instability and missing values. Therefore, a normalization strategy is adopted. LN is applied before the HSM-SSD layer for sequence stability, while BN is used in the DWConv and FFN layers to ensure training stability and improve inference efficiency.

The HSM-SSD module is the core component of EViM. It is designed to reduce the computational cost associated with linear projections in traditional SSD modules by compressing operations into a low-dimensional hidden state space. In the standard SSD [38] framework, the input sequence $X_{in}$ is first projected into a feature space to obtain the hidden state *h*, which is then processed through linear transformations and gating functions to generate the output. HSM-SSD shifts these high-cost operations from the feature space to a lower-dimensional hidden state space, significantly reducing computational complexity. Its structure is shown in Figure 2b, and the computation involves four main stages. First, the input sequence $X_{in}$ is linearly projected to generate an intermediate representation $\hat{B}$, a gating vector c, and positional encoding Δ. Local features are then extracted via DWConv, and a discretization module computes the weight matrix required for hidden state mapping, as shown in Equations (Equation 1) and (Equation 2):(1)$$\hat{B},\;c,\;\Delta =\mathrm{Linear}\left(X_{in}\right)$$(2)$$B,\;c=\mathrm{DWConv}\left(\hat{B},\;c\right),\;A,\;B=\mathrm{Discretization}\left(\hat{a},\;\hat{B},\;\Delta \right)$$
where $\hat{B}$, c, and Δ are obtained from the same input through independent linear layers, while the variable $\hat{a}\in R$ in Equation (Equation 2) is treated as a learnable scalar that controls the dynamics of the module.

Next, the input sequence is projected into a hidden state space to obtain a matrix $H_{\mathrm{in}}$ of size $R^{N\times D}$, where N≪L, effectively reducing the computational scale. Here, *L* represents the number of tokens, *N* denotes the number of states, and *D* refers to the number of channels. In the HSM-SSD layer, the input sequence of length *L* is compressed into *N* hidden states. The channel projection is then applied to these *N* states instead of *L* tokens, reducing the cost from $O(LD^{2})$ to $O(ND^{2})$ [26]. Therefore, when N≪L, the computation is significantly reduced. The equation is given as Equation (Equation 3):(3)$$H_{in}={\left(A⊙B\right)}^{T}X_{in}$$
where ⊙ is element-wise multiplication. Then, nonlinear feature mixing is performed in this compressed space using a gating mechanism and linear transformation, as expressed in Equation (Equation 4):(4)$$h,\;z=\mathrm{Linear}\left(H_{in}\right),\;h=\mathrm{Linear}\left(h⊙\sigma \left(z\right)\right)$$
where σ(·) denotes a sigmoid activation function.

Finally, the updated hidden state is projected back to the original feature space using the mapping matrix C to obtain the output sequence $X_{out}$ as shown in Equation (Equation 5):(5)$$X_{out}=C\;h$$

### 3.3. Loss Functions

A carefully designed loss function is crucial for abdominal multi-organ segmentation, as it not only guides the network to address the class imbalance among different organs and the blurred organ boundaries in images, but also directly determines the quantitative accuracy and clinical practical value of segmentation masks. This study adopts a weighted combination of pixel-level and regional-level losses as the objective loss function, defined as follows:(6)$$L_{\mathrm{total}}=w_{p}\cdot L_{p}+w_{r}\cdot L_{r}$$
where $L_{p}$ and $L_{r}$ denote the pixel-level and regional-level losses respectively, and $w_{p}$, $w_{r}$ are the corresponding weights with $w_{p}+w_{r}=1$.

Among the two components of this objective loss function, the pixel-level loss $L_{p}$ is specifically chosen as the Cross-Entropy (CE) loss, which is highly compatible with the core task requirements of abdominal multi-organ segmentation. In the abdominal multi-organ segmentation task, the CE loss can accurately match the requirement of predicting the category probabilities of each pixel for multiple organ categories. Its calculation formula is defined as Equation (Equation 7):(7)$$L_{CE}=-\frac{1}{N}\sum_{n=1}^{N}\sum_{c=1}^{C}t_{n}^{c}\log\left(y_{n}^{c}\right)$$
where *N* denotes the total number of pixels and *C* is the number of organ classes in the dataset, where C=8 in this study. Specifically, $t_{n}^{c}\in \left\{0,1\right\}$ represents the ground truth label of pixel *n* for class *c*, and $y_{n}^{c}\in \left[0,1\right]$ represents the predicted probability that pixel *n* belongs to class *c*.

At the regional level, the most frequently used overlap-based criteria in medical image segmentation are Dice, Jaccard, and Tversky losses [39]. Dice Loss is derived from the Dice coefficient, which measures the similarity between two sets and evaluates the overlap between the predicted and ground truth masks. The formula is given in Equation (Equation 8):(8)$$L_{\mathrm{dice}}=1-\frac{1}{C}\sum_{c=0}^{C-1}\frac{2\sum_{n=1}^{N}t_{n}^{c}y_{n}^{c}}{\sum_{n=1}^{N}\left(t_{n}^{c}+y_{n}^{c}\right)}$$

Dice loss emphasizes small structures and mitigates class imbalance.

Jaccard Loss is defined as the ratio of the intersection to the union of the predicted and ground truth masks. Similar to the Dice coefficient, it is computed per class and is formulated as Equation (Equation 9):(9)$$L_{\mathrm{jac}}=1-\frac{1}{C}\sum_{c=0}^{C-1}\frac{\sum_{n=1}^{N}t_{n}^{c}y_{n}^{c}}{\sum_{n=1}^{N}\left(t_{n}^{c}+y_{n}^{c}-t_{n}^{c}y_{n}^{c}\right)}$$

This loss imposes a stronger penalty on non-overlapping regions, which helps improve segmentation accuracy at the regional level.

Tversky Loss is a generalized form of Dice loss that introduces two hyperparameters, α and β, to control the penalties for false positives and false negatives, respectively. It is defined as Equation (Equation 10):(10)$$L_{tv}=1-\frac{1}{C}\sum_{c=0}^{C-1}\frac{\sum_{n=1}^{N}t_{n}^{c}y_{n}^{c}}{\sum_{n=1}^{N}\left(t_{n}^{c}y_{n}^{c}+\alpha t_{n}^{c}\left(1-y_{n}^{c}\right)+\beta y_{n}^{c}\left(1-t_{n}^{c}\right)\right)}$$
where α+β=1. A commonly used setting is α=0.4 and β=0.6, which enhances the sensitivity to small organ structures.

To determine the most appropriate regional-level loss for multi-organ abdominal segmentation, we conducted a systematic comparison; the quantitative results are presented in Section 4.4, where the optimal weights are also established through ablation studies. The systematic comparison and ablation studies determined that the optimal loss function has the form of Equation (Equation 11):(11)$$L_{\mathrm{total}}=0.4\cdot L_{CE}+0.6\cdot L_{jac}$$
where $L_{CE}$ and $L_{jac}$ denote the Cross-Entropy loss and Jaccard loss defined in Equation (Equation 7) and Equation (Equation 9), respectively.

## 4. Experiments

### 4.1. Datasets

We evaluate the proposed model on the Synapse [40] multi-organ CT dataset and the ACDC [41] cardiac MRI dataset. The Synapse dataset includes 30 abdominal CT scans with a total of 3779 contrast-enhanced axial slices. It is randomly divided into 2212 training slices and 1576 testing slices. The segmentation task focuses on eight abdominal organs: the aorta, gallbladder, left kidney, right kidney, liver, pancreas, spleen, and stomach. Each CT volume consists of 85 to 198 slices with a resolution of 512×512 pixels, and the voxel spacing ranges from $\left[0.54∼0.54\right]\times \left[0.98∼0.98\right]\times \left[2.5∼5.0\right]\;{\mathrm{mm}}^{3}$.

The ACDC dataset contains 100 cardiac MRI scans collected from various patients. All images were acquired during breath-hold to minimize motion artifacts and consist of sequential short-axis slices that cover the entire heart, from the base to the apex of the left ventricle. The slice thickness varies between 5 mm and 8 mm, while the in-plane spatial resolution ranges from 0.83 ${\mathrm{mm}}^{2}$ to 1.75 ${\mathrm{mm}}^{2}$ per pixel. Each scan is annotated with three cardiac structures: the right ventricle (RV), myocardium (Myo), and left ventricle (LV). In accordance with the TransUNet experimental setup, the dataset is split into 1903 training slices and 541 testing slices.

### 4.2. Implementation Details and Evaluation

All experiments are implemented using the PyTorch 2.0.0 framework and trained on an NVIDIA^®^ GeForce RTX^®^ 4090 D GPU (NVIDIA, Santa Clara, CA, USA) with 24 GB of memory. For both the Synapse and ACDC datasets, all images are resized to 224×224 pixels. To mitigate overfitting and enhance generalization, several data augmentation strategies are employed, including horizontal and vertical flipping, random rotation, Gaussian noise injection, and contrast adjustment. The model is trained for up to 300 epochs with a batch size of 12 using the SGD optimizer. The initial learning rate is set to 0.01, with a momentum of 0.9 and a weight decay of $1\times 10^{-4}$. For fair comparison, the number of training iterations is fixed at 20 k for the ACDC dataset and 14 k for the Synapse dataset.

Following commonly adopted evaluation metrics for segmentation tasks, we use the Dice Similarity Coefficient (DSC) and the 95th percentile Hausdorff Distance (HD95) to assess the performance. The DSC is computed as Equation (Equation 12):(12)$$DSC\left(X,Y\right)=\frac{2\cdot |X\cap Y|}{\left|X|+|Y\right|}$$
where *X* and *Y* denote the predicted segmentation and the ground truth, respectively.

HD95 is calculated as the 95th percentile of all pointwise surface distances between the predicted and ground truth boundaries. This metric is derived from the classical Hausdorff Distance (HD), which is defined as Equation (Equation 13):(13)HD(X,Y)=max{h(X,Y),h(Y,X)}
where $h\left(X,Y\right)=\max_{x\in X}\min_{y\in Y}d\left(x,y\right),\;h\left(Y,X\right)=\max_{y\in Y}\min_{x\in X}d\left(x,y\right)$, and d(x,y) represents the Euclidean distance between points *x* and *y*. By using the 95th percentile instead of the maximum, HD95 reduces the influence of outlier points and provides a more robust evaluation of segmentation boundaries.

### 4.3. Comparisons with State-of-the-Arts (SOTA)

Our method is compared with CNN-based, Transformer-based, and Mamba-based approaches, and the main experiments are conducted on the Synapse dataset to verify the superiority of the proposed model.

Table 1 summarizes the quantitative evaluation results of DSC and HD95. Specifically, the third and fourth columns display the average Dice and HD95 values over eight abdominal organs, respectively, while the fifth to twelfth columns provide the Dice values of individual organs. The proposed method yielded an average DSC of 82.67% and HD95 of 16.36 mm. Compared to CNN-based models, our model improves the average DSC by at least 3.75% and reduces HD95 by at least 0.08 mm, demonstrating superior capability in spatial representation and boundary delineation. Compared to Transformer-based methods such as TransUNet, our approach achieves a 5.19% improvement in DSC and a 15.33 mm reduction in HD95, indicating that the proposed efficient encoder design achieves a better balance between global context modeling and local detail preservation. Compared with the Mamba-based VM-UNet, our model improves DSC by 1.59% and reduces HD95 by 2.85 mm, further confirming the effectiveness of incorporating the state space duality mechanism in enhancing segmentation performance. The model exhibits consistent performance across organs of varying size and shape. For smaller structures, such as the gallbladder and right kidney, the DSC reaches 69.90% and 83.26%, which are 0.49% and 0.50% higher than those of the second-best method, respectively. For larger organs, such as the stomach and liver, the DSC improves by 2.56% and 0.61%, respectively. The segmentation results for the left kidney, pancreas, and spleen also closely follow the top-performing models, with all performance differences less than 0.55%, indicating robustness across organ sizes and shapes. Overall, the proposed method achieves either the best or highly competitive results in both global metrics and per-organ segmentation accuracy, fully demonstrating the effectiveness and advantages of our model in complex multi-organ segmentation tasks.

To more effectively showcase the superiority of our model’s segmentation, we have provided visualizations of the segmentation outputs of FMD-TransUNet, TransUNet, DA-TransUNet, VM-UNet and ours as shown in Figure 3. Specifically, in the first row, the segmentation results of all eight abdominal organs are highly consistent with the ground truth. This is particularly evident for small and morphologically complex structures such as the gallbladder and pancreas, where the model accurately identifies boundaries and produces complete segmentation regions. For medium-sized organs such as the left and right kidneys, which may contain cavities or exhibit inhomogeneous intensity distributions, the predicted contours are also clear and intact. Even in liver regions with anatomical variations or segmentation discontinuities, the model maintains consistent and reliable predictions. In the second row, the segmentation performance for the spleen and stomach is especially notable. The model is able to precisely distinguish the boundaries between adjacent organs, effectively avoiding confusion with surrounding tissues. In contrast, competing methods such as TransUNet and DA-TransUNet often exhibit overlapping or misclassified regions between the left kidney and spleen. The third row highlights the model’s ability to segment the renal hilum and the interface between the liver and stomach. Our model produces the most accurate segmentation of the left renal hilum, with smooth boundaries and precise localization, closely matching the ground truth. Moreover, in the region of tissue adhesion between the liver and stomach, the model successfully identifies subtle intensity transitions and accurately separates the two organs. In the fourth row, which features large-volume organ segmentation, the overall performance of our method is comparable to that of other approaches. However, our model still shows advantages in boundary refinement and spatial consistency.

To verify the robustness of the proposed model, we evaluated the performance of our model using the ACDC dataset and compared its results against SOTA models. The ACDC task involves multi-target segmentation of cardiac structures, which shares fundamental characteristics with the multi-organ segmentation task addressed by our model. This commonality makes ACDC a suitable benchmark for assessing the generalizability of our approach, which is specifically designed to handle such segmentation challenges through global context modeling and feature discrimination. As shown in Table 2, the proposed method achieves the highest average DSC of 90.53%, outperforming existing approaches. Specifically, for the segmentation of three cardiac structures, our model achieves the best DSC scores of 88.76% for the RV, 87.60% for the MYO, and 95.22% for the LV. In addition, the average HD95 is 1.15 mm, which is lower than that of the comparison models, further demonstrating the superior fine-grained boundary prediction capability of our method. As illustrated in the visual results in Figure 4, the proposed model produces contours that closely match the ground truth across various cardiac structures. Notably, in anatomically complex regions or in cases with pathologically blurred boundaries, the model is still able to accurately distinguish adjacent tissues and effectively identify irregular or low-contrast structures. These results demonstrate that the proposed model can generalize well.

### 4.4. Ablation Studies

To evaluate the influence of various conditions on segmentation results, we performed ablation experiments on our model on the Synapse dataset. It mainly includes: (1) Regional Loss Selection, (2) Weights Balancing between Pixel and Regional losses, (3) Ablation Study of EViM Module Contributions.

#### 4.4.1. Regional Loss Selection

As described in Section 3.3, the total loss function is composed of a pixel-level cross-entropy loss $L_{CE}$ and a regional-level loss $L_{r}$, weighted by $w_{p}$ and $w_{r}$, respectively. To determine the most effective regional loss for abdominal multi-organ segmentation, we conducted a systematic comparison among three commonly used overlap-based losses: Dice loss, Tversky loss, and Jaccard loss. In this experiment, the weighting coefficients $w_{p}$ and $w_{r}$ were both set to 0.5.

We compared TransUNet [12], FMD-TransUNet [18], and the proposed model using the same loss function settings. As shown in Table 3, the choice of region-level loss function leads to notable differences in segmentation performance. For the three models, the maximum variation in average DSC reached 2.06%, 1.56%, and 1.66%, respectively, with similarly significant differences observed in HD95. Among the three loss functions, Jaccard Loss consistently achieved the best overall performance, followed by Tversky Loss, while Dice Loss showed relatively lower performance. Under identical training configurations, our model achieved an average DSC of 82.23% and an HD95 of 17.62 mm when using Jaccard Loss, which significantly outperformed the other loss settings and models. These results further highlight the advantage of Jaccard Loss, as it directly reflects the degree of overlap between the predicted and ground truth regions. Its optimization objective is highly correlated with segmentation performance, enabling the model to generate clearer and more accurate boundaries during training. Therefore, the appropriate selection and combination of loss functions is of great importance for improving the overall segmentation performance of the model.

In addition, we conducted a detailed analysis of the loss trends during training. Figure 5a–c illustrates the convergence behavior of the three loss functions throughout the training process. It can be observed that Dice Loss stabilizes at an early stage (around 150 epochs), resulting in a relatively flat convergence curve. This early plateau limits further performance improvement in later stages and partially explains its inferior segmentation performance. In contrast, both Tversky Loss and Jaccard Loss exhibit more typical and stable convergence behavior, gradually converging after approximately 200 epochs with smooth and consistent training curves. Based on these observations, the number of training epochs for all subsequent experiments was set to 300. Given that Jaccard Loss consistently achieved the best segmentation performance across all three models, we ultimately selected it as the $L_{r}$. It was combined with the $L_{CE}$ to construct the final total loss function used for model training.

#### 4.4.2. Weights Balancing Between Pixel and Regional Losses

To determine the optimal loss function weight configurations, we conducted a weight sensitivity comparison experiment. Specifically, the Jaccard loss weight $w_{r}$ was varied from 0.3 to 0.7, and the CE loss weight $w_{p}$ was adjusted accordingly from 0.7 to 0.3. All other hyperparameters were kept consistent with the training settings in this experiment.

As shown in Figure 6, the DSC and HD95 curves corresponding to CE and Jaccard losses across all weight configurations clearly demonstrate their impact on segmentation performance. Notably, when $w_{r}=0.6$ and $w_{p}=0.4$, the model achieved the highest average DSC of 82.67% under the Jaccard loss. The HD95 metric exhibited a similar trend: as $w_{r}$ increased from 0.4 to 0.6, the HD95 value under the Jaccard loss continued to decrease, indicating improved boundary accuracy. The minimum HD95 value of 16.36 mm was reached when $w_{r}=0.6$. However, further increasing $w_{r}$ to 0.7 resulted in a rise in the HD95 value, suggesting that an excessively high weight for the Jaccard loss may lead to overfitting.

Taking both segmentation accuracy and boundary consistency into consideration, the optimal performance is achieved when the loss weights are set to $w_{p}=0.4$ for cross-entropy loss and $w_{r}=0.6$ for Jaccard loss.

#### 4.4.3. Ablation Study of EViM Module Contributions

To further validate the effectiveness of the EViM module, we designed a set of module comparison experiments. All comparison modules were inserted immediately before the Transformer encoder for fair comparison. The compared configurations include: no module insertion, DA module, DA+ module, LPA module, and the proposed EViM module.

The quantitative results in Table 4 indicate that the proposed EViM module achieves the best average performance on the Synapse dataset. It obtains a DSC of 82.67% and reduces the HD95 to 16.36 mm, clearly outperforming the other comparison modules. Compared to the model without any enhancement module, EViM improves the DSC by 3.56% and reduces the HD95 by 10.53 mm, demonstrating clear performance gains. Even when compared to the improved DA+ module, EViM still achieves a 2.49% improvement in DSC and a 3.6 mm reduction in HD95.

Further visual analysis from Figure 7 clearly demonstrates the differences in segmentation performance among the various modules on representative abdominal multi-organ cases. In the first row, the EViM module significantly outperforms other modules by more accurately identifying the boundaries between adjacent organs, such as the narrow region between the stomach and gallbladder, highlighting its strong boundary sensitivity. In the second row, EViM successfully segments both the main region of the liver and its smaller peripheral structures—an instance of fine-grained segmentation capability not achieved by other attention mechanisms. For small organs with vague boundaries that are often overlooked, such as the gallbladder and kidneys, EViM also exhibits higher segmentation precision, accurately delineating internal cavities and external contours, as shown in the second and fourth rows. Moreover, in the third and fifth rows, EViM provides stable segmentation results for larger organs such as the stomach, liver, and spleen. It shows enhanced robustness under challenging conditions, including intensity inhomogeneity, anatomical adhesion, and indistinct boundaries.

Based on the aforementioned ablation studies, we further conducted a preliminary efficiency evaluation. The proposed model achieves 37.34 G FLOPs and 138.53 M parameters, which is only a 2.53% increase in FLOPs and an additional 5.90 M parameters compared to the baseline. Although the computational cost is slightly higher, our method achieves the best mean DSC and the second-best HD95 result. Overall, our model effectively balances accuracy and efficiency, attaining higher precision without a significant increase in computational cost.

## 5. Conclusions

This study proposed a novel Transformer–Mamba hybrid model that embeds an EViM module before the Transformer layers in the encoder to achieve efficient global context modeling and channel interaction. An improved loss function was further incorporated to enhance segmentation performance. Experimental results on the CT-based Synapse dataset demonstrated that the proposed model achieved a Dice score of 82.67% and an HD95 of 16.36 mm, outperforming current state-of-the-art methods. Generalization experiments on the MR-based ACDC dataset also achieved superior results, indicating that the method can deliver stable and high-accuracy segmentation across different imaging modalities and multi-target conditions. Our model still has certain limitations. Although the average DSC has reached an optimal level, the segmentation performance for some individual organs (such as the aorta and left kidney) is still not optimal, indicating that the model has room for improvement in fine-grained structure recognition and boundary feature modeling. Future work will focus on enhancing fine-grained feature representation and boundary delineation. Specifically, we plan to extend the current 2D framework to 3D to better model spatial context and volumetric relationships in medical scans. Additionally, integrating multi-modal data will be explored to provide complementary information and improve segmentation robustness, especially for organs with low contrast or complex shapes.

## Figures and Tables

**Figure 1 sensors-25-06785-f001:**
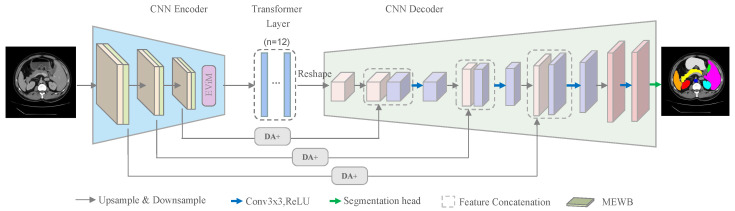
Overall Architecture of the Proposed Model.

**Figure 2 sensors-25-06785-f002:**
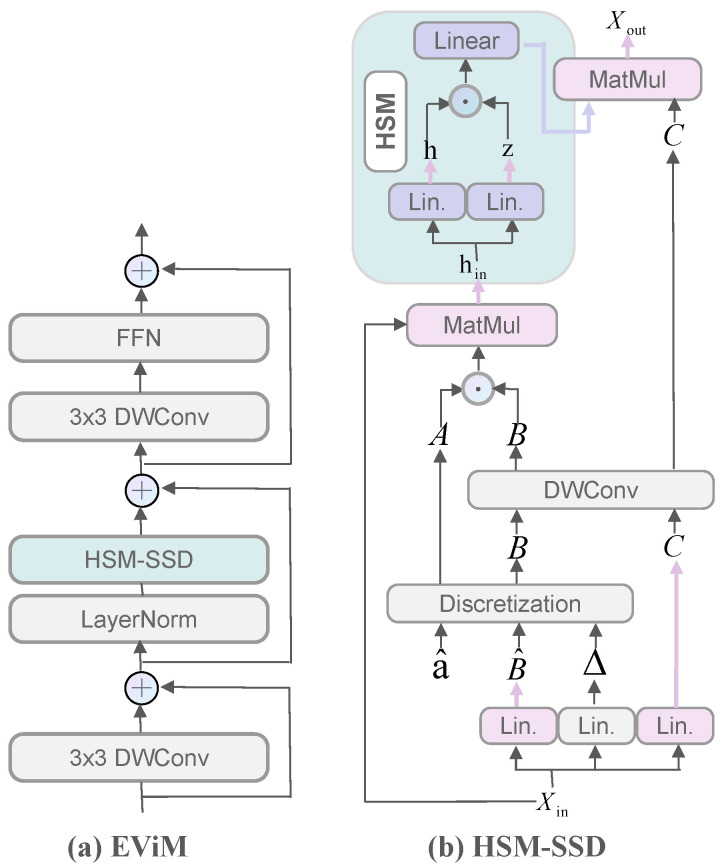
(**a**) Framework of the EViM Module. (**b**) Architecture of the HSM-SSD Mechanism (Figures adapted from [26] under the CC BY-SA 4.0 license).

**Figure 3 sensors-25-06785-f003:**
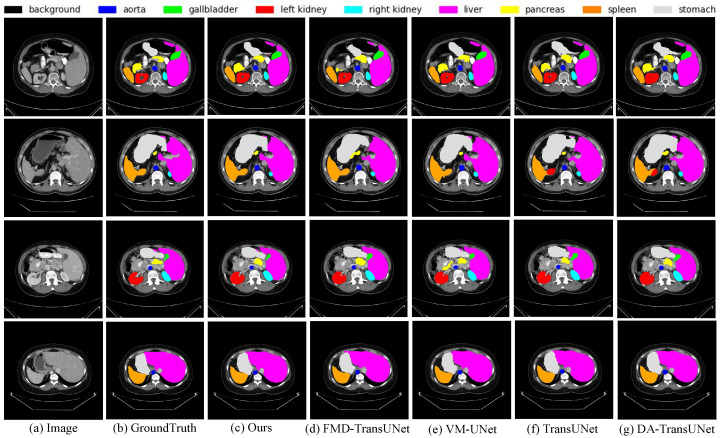
Qualitative Visualization of Segmentation Results on the Synapse Dataset.

**Figure 4 sensors-25-06785-f004:**
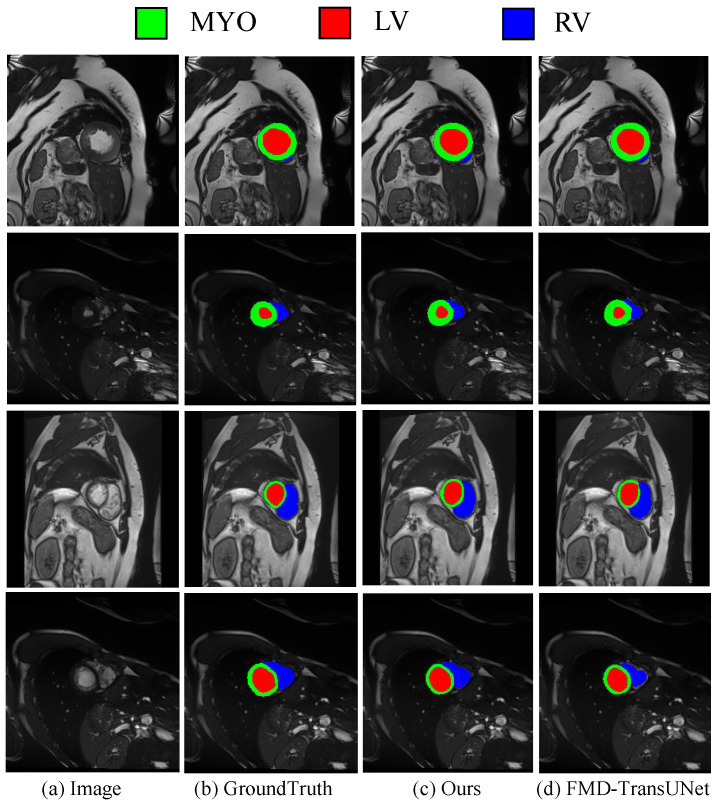
Qualitative Visualization of Segmentation Results on the ACDC Dataset.

**Figure 5 sensors-25-06785-f005:**
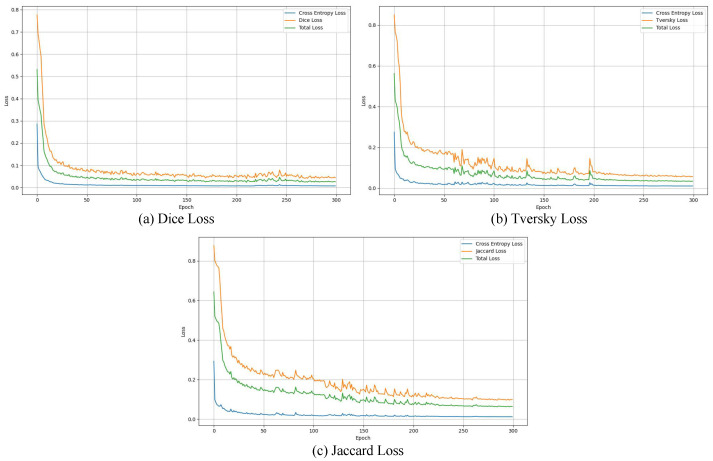
Comparison of Three Region-Level Loss Functions.

**Figure 6 sensors-25-06785-f006:**
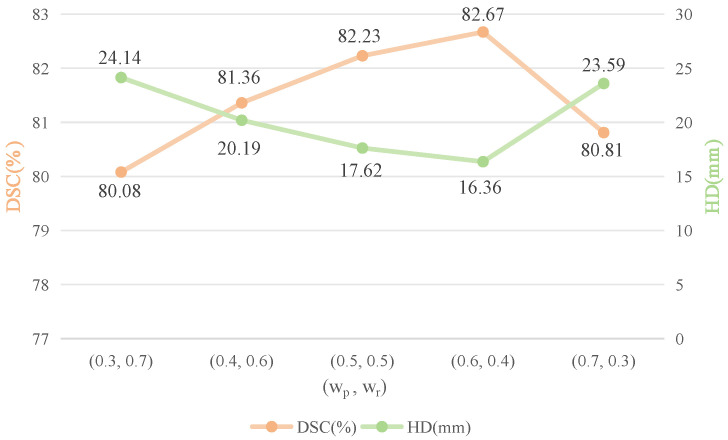
Trends of DSC and HD95 under different weight settings of CE and Jaccard loss.

**Figure 7 sensors-25-06785-f007:**
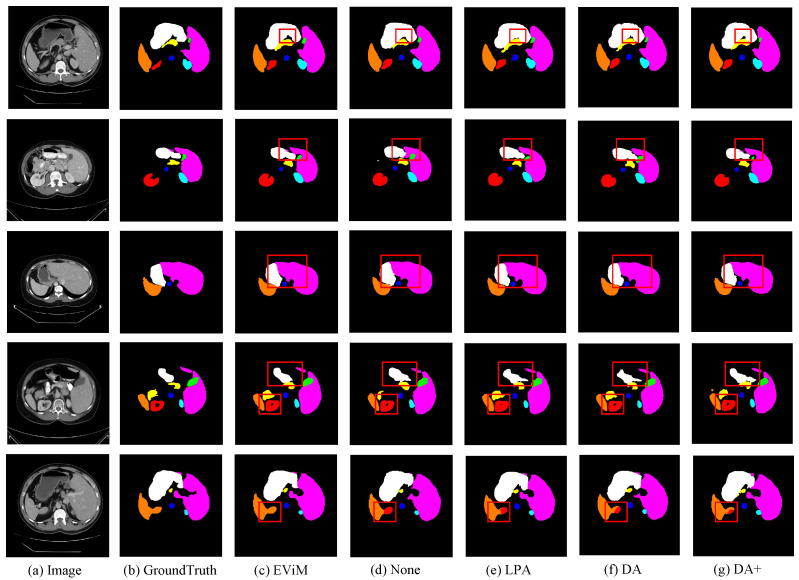
Visual Comparison of Different Modules Before the Transformer Layer in the Encoder.

**Table 1 sensors-25-06785-t001:** The quantitative results on the Synapse dataset, where the best scores are shown in bold and the second-best results are underlined.

Type	Method	HD95↓	DSC↑	Aorta	Gallbladder	Kidney (L)	Kidney (R)	Liver	Pancreas	Spleen	Stomach
CNN	UNet [4]	39.70	76.85	89.07	69.72	77.77	68.60	93.43	53.98	86.67	75.58
	UNet++ [5]	36.93	76.91	88.19	68.89	81.76	75.27	93.01	58.20	83.44	70.52
	Residual UNet [6]	38.44	76.95	87.06	66.05	83.43	76.83	93.99	51.86	85.25	70.13
	Attention UNet [7]	36.02	77.77	**89.55**	68.88	77.98	71.11	93.57	58.04	87.30	75.75
	MEW-UNet [9]	16.44	78.92	86.68	65.32	82.87	80.02	93.63	58.36	90.19	74.26
ViT	TransUNet [12]	31.69	77.48	87.23	63.13	81.87	77.02	94.08	55.86	85.08	75.62
	TransNorm [17]	30.25	78.40	86.23	65.10	82.18	78.63	94.22	55.34	89.50	76.01
	Swin-UNet [13]	21.55	79.13	85.47	66.53	83.28	79.61	94.29	56.58	**90.66**	76.60
	MT-UNet [10]	26.59	78.59	87.92	64.99	81.47	77.29	93.06	59.46	87.75	76.81
	DA-TransUNet [16]	23.48	79.80	86.54	65.27	81.70	80.45	94.57	61.62	88.53	79.73
	TransDeepLab [28]	21.25	80.16	86.04	69.16	84.08	79.88	93.53	61.19	89.00	78.40
	FMD-TransUNet [18]	**16.35**	81.32	88.76	65.23	85.12	82.12	94.19	**66.31**	89.73	79.13
Mamba	VM-UNet [23]	19.21	81.08	86.40	69.41	**86.16**	82.76	94.17	58.80	89.51	81.40
	Ours	16.36	**82.67**	88.26	**69.90**	84.49	**83.26**	**95.18**	65.76	90.52	**83.96**

**Table 2 sensors-25-06785-t002:** Quantitative comparison of our method and other approaches on the ACDC dataset. The best results are highlighted in bold, and the second-best results are underlined.

Type	Method	HD95↓ (mm)	DSC↑ (%)	RV	MYO	LV
CNN	R50-UNet [4]	-	87.60	84.62	84.52	93.68
	R50 AttnUNet [7]	-	86.90	83.27	84.33	93.53
	UNet++ [5]	-	89.58	87.23	87.13	94.37
	DeepLabv3+ [42]	-	88.25	85.41	85.44	93.90
ViT	ViT-CUP [11]	-	83.41	80.93	78.12	91.17
	R50 ViT [11]	-	86.19	82.51	83.01	93.05
	R50-ViT-CUP [11]	-	87.57	86.07	81.88	94.75
	TransUNet [12]	-	89.71	86.67	87.27	95.18
	Swin-UNet [13]	-	88.07	85.77	84.42	94.03
	UNETR [43]	-	86.61	85.29	86.52	94.02
	MT-UNET [10]	2.23	90.43	86.64	**89.04**	**95.62**
	FMD-TransUNet [18]	2.07	90.10	87.87	87.48	94.96
	Ours	**1.15**	**90.53**	**88.76**	87.60	95.22

**Table 3 sensors-25-06785-t003:** Performance of Dice, Tversky, and Jaccard Loss Functions on TransUNet, FMD-TransUNet, and Our method on the Synapse dataset.

Loss Function	Model	DSC (%)	HD95 (mm)
Dice Loss	TransUNet	77.48	31.69
	FMD-TransUNet	79.09	28.87
	Ours	80.57	21.01
Tversky Loss	TransUNet	79.09	28.87
	FMD-TransUNet	80.23	23.59
	Ours	80.29	20.64
Jaccard Loss	TransUNet	79.54	27.55
	FMD-TransUNet	81.65	17.98
	Ours	82.23	17.62

**Table 4 sensors-25-06785-t004:** Comparison of Different Modules Inserted Before the Transformer Layer in the Encoder.

Module	DSC (%)	HD95 (mm)
-	79.11	26.89
DA	79.82	23.06
DA+	80.18	19.96
LPA	79.60	24.01
EViM	82.67	16.36

## Data Availability

The datasets used in this study are publicly available. The Synapse dataset can be accessed at https://www.synapse.org/#!Synapse:syn3193805/wiki/217789 (accessed on 18 May 2024), and the ACDC dataset is available at https://www.creatis.insa-lyon.fr/Challenge/acdc/ (accessed on 20 August 2024).

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
