# Peer review of "An Efficient Vision Mamba–Transformer Hybrid Architecture for Abdominal Multi-Organ Image Segmentation"

_sensors, 2025, doi:10.3390/s25216785_

Round 1

Reviewer 1 Report

Comments and Suggestions for Authors

Please check the attachment

Reviewer 2 Report

Comments and Suggestions for Authors

comments on sensors-3917509
The authors of this manuscript employed a Transformer encoder with a EViM to improve the accuracy for multi-organ segmentation in the abdomen. The datasets employed in this study are CT-based Synapes and MRI-based ACDC. The results showed that the model combining the Transformaer encoder and an EViM module exhibited a improved segmentation for 8 organs in the abdomen. Although this manuscript is well organized, I would like to give my comments and suggestions as below for possible improvements for this paper.

1. There are less technical details for this study. I would encourage the authors to describe the necessary development details to reflect their contribution to this work. For example,

1.1. What were the reasons in the authors mind to place the EViM before the Transformer encoder. Why not place the EViM in other layer in their model in Fig. 1? 

1.2. As the authors announce they jointly captured the local-global features in line 200 on page 5, is this model in Fig. 1 the optimal structure for this purpose?

2. There are some parameters not disclosed. Please clarify what they were in this study. 

2.1. In the encoder in Sec. 3.1 at line 204 on page 5, what was the number D for the D-dimensional vector?
2.2. For the FFN layer from line 225 to line 227, please describe in details how the 1X1 convolution was implemented with the channel expansion ratio of 4? Is there a special meaning for the ratio by 4? If there is space, the authors can also describe possible effects if the ratio was other number rather than 4.
2.3. Even there are citations for both Layer Normalization (LN) and Batch Normalization (BN) in lines 222 and 223, it is better to announce how to do in this study. It is worthy to give more space to describe the effects of LN and BN on the stability of training and inference efficiency of the outputs in lines 229 and 230.
2.4. It is not clear how to make a linear mapping for ^B, C, Delta from xin in eq. (1). Please make the description more clear. Were the linear functions the same for ^B, C, Delta? There is similar issue for eq. (2) . What was ^a in eq.(2) and Fig. 2. How much is ^a and where does it come from?
2.5. What was the number L in line 244? and why it will reduce the computation scale for N << L?
2.6. The total loss function was derived from pixel-level loss and regional-level loss, Lp and Lr. Please describe the formulas for Lp and Lr for clarity.
2.7. In eq.(7), I think C in line 267 would be 8 for this study. I think the authors can make it more clear to express C=8 to make connection between Sec. 3 and the results in Fig. 3.
2.8. It seemed that the dataset ACDC was not employed for the multi-organ segmentation in this study. As Table 2 and Fig. 4 shows, only heart was investigated. The authors need to explain how the proposed model would help in segmentation of a single organ.
I hope there would be possible improvement to make this paper more clear to show the contribution of the authors in this study.

Round 2

Reviewer 1 Report

Comments and Suggestions for Authors

The paper can be accepted in its current form
